# Brain glucose and ketone metabolism in first-episode psychosis: Neuroimaging and brain metabolism before and after antipsychotic treatment: The protocol for the CAST-ATP study

Kevin Zemmour [1,2‡]*, Guy-Olivier Samson[1‡], Mélanie Fortier[2], Eliot Parent[1], Alexandra Leus[3], Sylvain Grignon[1], Jean-Daniel Carrier [1], Kevin Whittingstall[4], Andreas Aalkjær Danielsen[5,6], Ole Köhler-Forsberg[5,6], Allan Kjeldsen Hansen[6,7], Sri Mahavir Agarwal[8,9,10], Anna Cristina Andreazza[8,9,10,11,12], Bjørn Hylsebeck Ebdrup [13,14], Margaret Hahn [8,9,10,11,14‡]*, Stephen Cunnane[2,15‡]*

1 Department of Psychiatry, Faculty of Medicine and Health Sciences, University of Sherbrooke, Sherbrooke, Quebec, Canada, 2 Research Center on Aging, University of Sherbrooke, Sherbrooke, Quebec, Canada, 3 Community Members, 4 Department of Diagnostic Radiology, University of Sherbrooke, Sherbrooke, Quebec, Canada, 5 Psychosis Research Unit, Aarhus University Hospital Psychiatry, Aarhus, Denmark, 6 Department of Clinical Medicine, Aarhus University, Aarhus, Denmark, 7 Department of Nuclear Medicine & PET, Aarhus University Hospital, Aarhus, Denmark, 8 Department of Pharmacology & Toxicology, University of Toronto, Toronto, Ontario, Canada, 9 Institute of Medical Sciences, University of Toronto, Toronto, Ontario, Canada, 10 Department of Psychiatry, University of Toronto, Toronto, Ontario, Canada, 11 Centre for Addiction and Mental Health, Toronto, Ontario, Canada, 12 Mitochondrial Innovation Initiative, MITO2i, University of Toronto, Ontario, Canada, 13 Center for Neuropsychiatric Schizophrenia Research (CNSR), Mental Health Center, Glostrup, Copenhagen University Hospital – Mental Health Services CPH, Copenhagen, Denmark, 14 Department of Clinical Medicine, Faculty of Health and Medical Sciences, University of Copenhagen, Copenhagen, Denmark, 15 Division of Endocrinology, Department of Medicine, University of Sherbrooke, Quebec, Canada

‡ KZ and GOS share first authorship on this work. MH and SC are joint senior authors on this work.
* Margaret.Hahn@camh.ca (MH); Stephen.Cunnane@USherbrooke.ca (SCC); kevin.zemmour@usherbrooke.ca (KZ)

## Abstract

First episode of psychosis (FEP) has an early onset and is associated with significant functional impairment, loss of productivity and premature cardiovascular disease. Antipsychotics (AP) remain the cornerstone treatment of FEP yet they fail to improve key symptom domains and contribute to the metabolic burden of this disorder. A growing body of evidence suggests that a metabolic deficit in the brain, specifically of glucose, at the earliest stages of illness could represent an etiopathological phenotype of FEP. Correcting this metabolic deficit could improve outcomes and disease course. The acronym for this study is CAST-ATP for the collaboration between our clinical research sites in *C*openhagen, *A*arhus, *S*herbrooke and *T*oronto, on the subject of *A*ntipsychotic (AP) treatment, PE*T* and *P*sychosis. The main aims of CAST-ATP are to evaluate the effect of 1) a diagnosis of FEP, and, 2) 4–6 weeks of AP treatment on brain energy metabolism measured by PET scans (uptake of ketones and glucose). The hypothesis is that (i) glucose metabolism will be impaired

which permits unrestricted use, distribution, and reproduction in any medium, provided the original author and source are credited.

**Data availability statement:** No datasets were generated or analysed during the current study. All relevant data from this study will be made available upon study completion.

**Funding:** Financial support for this article and the project it describes was received in the form of a philanthropic grant donated by the "Baszucki Brain Research Fund", USA, to the Université of Sherbrooke Foundation. SCC holds the Endowed Clinical Ketotherapeutics Chair at the Université of Sherbrooke. MH holds the CAMH and UofT Meighen Family Research Chair in Psychosis Prevention. SMA is supported in part by the Discovery Fund, CAMH and the Academic Scholars Award from the Department of Psychiatry, University of Toronto. The funders had no role in study design, data collection and analysis, decision to publish, or preparation of the manuscript.

**Competing interests:** SCC has received research funding and test materials from Nestlé Health Science. He consults for Nestlé Health Science and Cerecin. OKF reports speaker fees from Lundbeck Pharma A/S, consultant work for WCG International, and serving on an advisory board for Boehringer-Ingelheim. BE is part of the Advisory Board of Boehringer Ingelheim, Lundbeck Pharma A/S; and has received lecture fees from Boehringer Ingelheim, Otsuka Pharma Scandinavia AB, and Lundbeck Pharma A/S. SMA has received honoraria from HLS therapeutics and Boehringer Ingelheim. MH has received consultant fees from Alkermes and MERCK. The other authors declare that the research will be conducted in the absence of any commercial or financial relationships that could be construed as a potential conflict of interest. This does not alter our adherence to PLOS ONE policies on sharing data and materials.

**Abbreviations:** [$^{11}$C]AcAc: 1-[$^{11}$C]acetoacetate; AP: Antipsychotics; [$^{18}$F]FDG: 2-[$^{18}$F]fluoro-2-deoxy-D-glucose; FEP: First episode of psychosis; [$^{11}$C]OHB: R-β-[1-$^{11}$C]hydroxybutyrate; PET: Positron emission tomography; MRI: Magnetic resonance imaging.

in AP-naïve patients as compared to healthy controls, and (ii) this defect will be worsened by AP. In contrast, across the two aims, brain ketone metabolism is predicted to not be significantly influenced by FEP or AP treatment. Participants on both sites will undergo an imaging protocol (PET scans + MRI) in addition to measures of psychopathology and related peripheral metabolic, inflammatory and hormonal markers. If our hypothesis is confirmed, it will reinforce the strategy to leverage ketone supplementation to improve symptoms, functioning and quality of life by bypassing the brain glucose deficit in FEP. As such, this should be a significant therapeutic development. To this last point, the pharmaceutical treatment of schizophrenia spectrum disorders has not progressed beyond currently available AP for over seven decades.

## Introduction

Patients experiencing a first episode of psychosis (FEP) face a poor prognosis if not promptly and effectively managed and if relapses are not prevented [1,2]. Chronic psychotic and mood symptoms [3,4], reduced treatment response [5,6], and atrophy of some brain regions [7] are commonly noted in FEP if the disorder becomes chronic. Ultimately, there is a decrease in quality of life and a loss of 10–20 years of life expectancy, largely attributable to premature cardiovascular disease [8,9].

Clinics specializing in FEP aim to treat this syndrome as early as possible and prevent or rapidly address relapses so as to reduce the burden for the individual, family and society. While second-generation antipsychotics (AP) remain the gold standard treatment for FEP, they are frequently associated with significant metabolic side effects which negatively impact metabolic comorbidity, in addition to medication adherence, and quality of life [10].

Disrupted brain energy metabolism has been proposed as a risk factor for developing psychosis [11,12] as well as mood disorders [13,14]. Adenosine triphosphate (ATP), the primary energy currency of cells, is produced through various metabolic pathways involving the oxidation of glucose, fatty acids or ketone bodies. Glucose is the main substrate for ATP production in the brain [15]. Increasing evidence suggests that impaired glucose metabolism is associated with declining brain function and symptoms in mental health disorders [16]. Ketone bodies – β-hydroxybutyrate (OHB) and acetoacetate (AcAc) – are the brain's second most important fuel. Unlike glucose, ketones are not converted into lactate. Rather, they generate ATP exclusively through mitochondrial oxidative phosphorylation. Increased availability of ketones is hypothesized to potentially bypass the brain energy gap caused by defective brain glucose metabolism in bipolar disorder and psychosis [12,13]. Ketones can be generated endogenously or consumed pre-formed as exogenous supplements. Endogenous ketone production occurs primarily in the liver during conditions which result in lower insulin, such as fasting or very low carbohydrate intake. Exogenous ketone supplements are available as medium-chain triglycerides, ketone salts, and ketone esters. Ketogenic diets and exogenous ketones have shown promise in managing Alzheimer's disease and mild cognitive impairments [17,18] and have long

been known to be effective in refractory epilepsy [19,20]. In the case of Alzheimer disease, unlike with the brain's energy deficit in glucose, ketone uptake and utilization by the brain is not impaired, so more ketones in blood directly translate to increased brain ketone metabolism and a smaller brain energy gap [21]. This 'brain energy rescue' therapeutic strategy has prompted the exploration of a ketogenic diet as an novel therapeutic intervention for bipolar disorder and schizophrenia [14,22–25]. As an alternative approach, exogenous ketogenic supplements are also undergoing testing for these disorders [14,26].

Despite these encouraging clinical reports, to our knowledge, there has been no direct measurement of brain glucose and ketone metabolism in FEP or other forms of severe mental illness. Moreover, the effect of AP on brain energy metabolism has not been studied despite their well-established metabolic adverse effects including cardiometabolic morbidity, worse cognition, poorer quality of life, and decreased medication adherence [16,27–31]. In individuals with schizophrenia, aripiprazole and ziprasidone are reported to be associated with a reduction in peripheral ketones in by approximately 50%, bringing values back to control levels [32]. However, there is no direct evidence that: i) Ketone utilization could be enhanced by antipsychotics, in a disorder where glucose utilization is impaired [33] and/or ii) whether ketones continue to be used normally by the brain in FEP and/or they could bypass a deficit in brain glucose utilization.

On both our sites (Aarhus, Denmark, and Sherbrooke, Québec, Canada), we have been routinely assessing glucose and ketone metabolism by PET. The Sherbrooke site has accumulated over 400 such dual tracer brain scans in young adults, older adults, Alzheimer disease and Parkinson disease, before and after ketogenic interventions [21].

We describe here the protocol for the CAST-ATP observational study (*C*openhagen, *A*arhus, *S*herbrooke and *T*oronto - *A*nti-psychotics, *P*ET and *P*sychosis study, a collaboration) in which both brain glucose and ketone metabolism will be quantified in young adults with FEP both before and 4–6 weeks after AP initiation.

## Materials and methods

### Study design: Overview, hypothesis and objectives

The Sherbrooke site protocol is registered on ClinicalTrials.gov (NCT06651112) and is approved by the Research Ethics Committee of the Centre Intégré Universitaire de santé et de services sociaux of Estrie region, Quebec, Canada (2025−5589). The Aarhus site protocol is currently preapproved by ethics committee. The radiotracer for glucose is 2-[$^{18}$F]fluoro-2-deoxy-D-glucose ([$^{18}$F]FDG) on both sites, while for ketones it will be 1-[$^{11}$C]acetoacetate ([$^{11}$C]AcAc) in Sherbrooke and R-β-[1-$^{11}$C]hydroxybutyrate ([$^{11}$C]OHB) in Aarhus. We have previously demonstrated that the two tracers are equivalent for analyzing ketone metabolism [34].

The hypotheses are that: (i) lower global and regional glucose uptake will be observed in AP-naïve FEP compared to matched healthy controls, a difference that will be exacerbated 4–6 weeks after the initiation of AP, and brain ketone metabolism will not be significantly changed in FEP compared to healthy controls, independently of 4–6 weeks post-initiation of AP treatment. (ii) Lower brain glucose uptake will be directly correlated with plasma lactate as an indirect measure of mitochondrial dysfunction, and with impaired peripheral glucose metabolism and cognitive performance before and after AP initiation.

Primary Objective: To compare brain uptake of glucose and ketones between AP-naïve FEP and healthy controls (matched for age and sex), before and 4–6 weeks after AP initiation.

Secondary Objective: In FEP, to compare the measures of brain energy metabolism to clinical improvement over time, as measured by the percentage change in the Brief Psychiatric Rating Scale or Positive and Negative Syndrome Scale (BPRS).

Exploratory Objectives: (i) To examine associations in relation to brain energy metabolism to other measures of psychopathology (e.g., cognition, depressive symptoms), metabolic state (weight, lipids, inflammation markers, systemic glucose metabolism as measured by 10 days of continuous glucose monitoring (CGM)), and brain structure and

cerebro-vasculature. (ii) To assess whether CGM data are a valid proxy for brain FDG results. (iii) To assess the acceptability and feasibility of CGM in FEP.

## Similarities and differences on the two sites (Table 1)

### Recruitment and screening

**Target population.** In the FEP program at CIUSSS de l'Estrie-CHUS (Sherbrooke, Québec) and in the first episode of schizophrenia program (FES, a sub-group of FEP. When relevant, FES will be distinguished from FEP; otherwise, the term FEP will be used throughout this article to encompass both FEP and FES.) of the Central Denmark Region Schizophrenia Cohort (Aarhus, Denmark), participants between 18 and 35 years old, newly admitted to the FEP program, AP-naïve, who wish to start an AP after 10 days of delay, will be invited to participate.

**Sample size.** We are not aware of any existing work measuring brain ketone metabolism before or after AP which we could be used to estimate an appropriate sample size in this population. Our previous studies on brain ketone and FDG PET showed group differences with sample sizes of 7–15 participants [17,34,35]. These previous results support recruitment of a total of 36 participants with FEP (18 per site). The principal features of the protocols for our two sites are shown in Table 1.

**Inclusion criteria.** FEP will be diagnosed by a psychiatrist based on the Diagnostic and Statistical Manual of Mental Disorders, 5th ed. (DSM-5-TR) criteria (Sherbrooke), while FES will be diagnosed according to the International Classification of Diseases 10th Revision (ICD-10) criteria (Aarhus). They will be either outpatients or inpatients who are willing to begin taking an AP (regardless of change in drug and/ or dose during the study). Patients will need to be able to read, understand and express themselves in French or English (Sherbrooke), or Danish or English (Aarhus), and must be capable of understanding and signing consent.

**Exclusion criteria.** Participants will be excluded if they are currently on an AP (see exceptions below), or if there is an history of AP use for morethan 12 weeks or more in their lifetime, with a current wash-out period of 2 weeks, prior to

**Table 1. Overview of similarities and differences in the two sites.**

| | Sherbrooke | Aarhus | Comments |
|---|---|---|---|
| **Continent** | North America | Europe | Analysis of two different cities, whose lifestyle and food cultures differ, but whose patient care is similar. Thus, if we observe similar results, they will have a more general scope and independent of the lifestyle and eating habits. |
| **Target population** | First episode of psychosis (FEP) | First episode of schizophrenia (FES) | FEP include all first episodes of psychosis regardless of the outcome of the diagnosis, whether schizophrenia spectrum disorder, mood disorders, or personality disorders. FES, a sub-group of FEP, includes only those that will develop into schizophrenia spectrum disorders. Having wider selection criteria in Sherbrooke but narrower in Aarhus will make it possible to better analyze whether FEP or FES is associated with a greater or lesser brain metabolic deficit. |
| **Radiotracers for PET** | [$^{18}$F]FDG [$^{11}$C]AcAc | [$^{18}$F]FDG [$^{11}$C]OHB [$^{15}$O]H$_2$O | [$^{11}$C]AcAc and [$^{11}$C]OHB are equivalent for analyzing ketone metabolism [34]. [$^{15}$O]H$_2$O is the gold standard for measuring tissue perfusion. |
| **Participants (n)** | 18 with 10 re-scans | 18 with 12 re-scans | |
| **Healthy controls (n)** | Matched healthy controls from our database | 12 newly recruited healthy controls | |

*FEP – first episode of psychosis, FES – first episode of schizophrenia, PET scan – positron emission tomography scan, [$^{11}$C]AcAc – 1-[$^{11}$C]acetoacetate, [$^{18}$F]FDG – 2-[$^{18}$F]fluoro-2-deoxy-D-glucose, [$^{11}$C]OH – R-β-[1-$^{11}$C]hydroxybutyrate.*

study start, with the exception of injectables, which are excluded. The exception will be aripiprazole if it is taken at less than 2.5 mg/day or quetiapine at less than 50 mg/day, regardless of duration or timing of the prescription. These two exclusions will help relieve symptoms of anxiety, distress and insomnia (and, hence, recruitment) with a minimal predicted impact on brain energy metabolism. The following comorbidities will lead to exclusion on both sites: known intellectual disability, autism spectrum disorder, moderate to severe substance use disorder, psychosis induced by a medical condition, psychosis induced by drug use or withdrawal, acute suicidal ideation, diabetes mellitus, and other conditions that could interfere with participation according to the judgment of the qualified physician. These criteria are assessed by the psychiatrist according to review of the patient's file, the psychiatric interview, and during the first eligibility visit by a member of the research team. Presence of a metallic object in the body that is incompatible with MRI, or pregnancy, childbirth in the last 6 months or breastfeeding that is incompatible with radiotracer injection will also result in exclusion.

### Recruitment

Only FEP patients for whom the participating psychiatrist deems it appropriate and safe to undergo tests requiring a ≤ 10-day delay before starting an AP will be invited to participate. A member of the research team (researcher or research professional) will then contact the person to present the project and obtain their written consent.

Participant recruitment began in 04 October 2024 and is expected to be completed by 04 February 2026. Considering the additional two months planned for patient participation, data collection and analysis should be finalized by spring 2026.

### Study overview

After the first visit to the FEP clinic, patients whom a psychiatrist deems appropriate and safe to participate will be invited to the V0-Eligibility visit, at which time the participant will be informed about the study. Informed consent, eligibility criteria, sociodemographic data, medication list, MRI contraindication, and the subsequent visits are done during this visit (Fig 1 and Fig 2).

During V1- and V3- Clinic visits (before and after AP), the research nurse will obtain a 12 h fasting blood sample then will measure anthropometrics and vital signs. Face-to-face interviews and self-reported questionnaires with the research team will be completed (Fig 1). Before the V2- and V4- Imaging visits (before and after AP) the participant will need to have fasted for at least 4 hours. The research team will reconfirm consent and complete questionnaires on akathisia symptoms and food craving (Fig 1). A pregnancy test will be repeated for women. The Imaging visits involve MRI and PET scans. Optional CGM (Dexcom G7) will be proposed right after the Imaging visits. Compatibility of the CGM device with the participant's mobile phone (to be supplied if necessary) will be verified to upload the data and instructions provided (synchronization period, precautions for bathing, etc.). The CGM device will be removed 8–10 days later.

### Outcome measures

Data will be collected (Fig 1) during face-to-face interviews and be performed by research team members in psychiatric outpatient's or research center's offices depending on the most convenient option for participants. Participants will be invited to complete self-reported questionnaires independently or to ask for assistance for all or part of the questionnaires.

### Brain ketone and glucose metabolism: PET scanner, MRI procedure and image analysis

The Sherbrooke site will perform all PET scans on a Biograph Vision 600 scanner (Siemens, Erlangen, Germany) with a 26-cm axial field of view. The image-derived input function will be cross-calibrated against the plasma radioactivity counted in a gamma counter (Cobra, Packard, USA) of blood samples acquired during each PET session (Castellano 2017, Croteau 2018). [$^{11}$C]AcAc is used to visualize brain ketone uptake and the site has expertise in multi-imaging approaches, combining multi-tracer PET with various MRI modalities (volumetric, functional, diffusion, etc.) [17,36–39]. FEP data will be compared to an existing database of healthy controls from Sherbrooke.

| | Enrolment | BASELINE | | POST | |
|---|---|---|---|---|---|
| **Timepoint (weeks)** | T0 | 0-1 | 0-1 | 5-7 | 5-7 |
| **ENROLMENT** | | | | | |
| Eligibility screen | X | | | | |
| Informed consent | X | | | | |
| **METABOLIC ASSESSMENTS** | | | | | |
| Blood sample | | X | | X | |
| Physical measures (Weight, height, waist circumference, blood pressure) | | X | | X | |
| Continuous glucose monitoring sensor placement | | | X | | X |
| Pregnancy test when applicable | X | | X | | X |
| **PSYCHIATRIC ASSESSMENTS** | | | | | |
| Duration of untreated illness assessment | X | X | | | |
| Brief Psychiatric Rating Scale (BPRS) | | X | | X | |
| Clinical Global Impression, Severity (CGI-S) | | X | | X | |
| Calgary Depression Scale for Schizophrenia (CDSS) | | X | | X | |
| Global Assessment of Functioning (GAF) | | X | | X | |
| Brief Assessment of Cognition (BACS) | | X | | X | |
| Side Effect Rating Scale (UKU) | | X | | X | |
| Alcohol Use Disorders Identification Test (AUDIT) | | X | | X | |
| Drug Disorders Identification Test (DUDIT Drug) | | X | | X | |
| Fagerström Test for Nicotine Dependence (FTND) | | X | | X | |
| Epsworth Sleepiness Scale (ESS) | | X | | X | |
| Simple Physical Activity-questionnaire (SIMPAQ) | | X | | X | |
| Barnes akathisia scale (BAS) | | | X | | X |
| Food Craving questionnaire (FCQ) + Visual analogue scale | | | X | | X |
| Medication list (type + dose) | X | | | X | |
| Medication Adherence Rating Scale (MARS) | | | | X | |
| **BRAIN GLUCOSE AND KETONES ASSESSMENTS** | | | | | |
| Brain imaging PET scan | | | X | | X |
| Brain imaging MRI scan | | | X | | |

**Fig 1. Overview schedule of enrolment, and assessments in Canada.** V – visit, PET scan – positron emission tomography scan, MRI scan – magnetic resonance imaging scan.

PET scans will be matched to brain regions by an MRI acquired on the same day (3 Tesla with a 32-channel head coil, Ingenia, Philips Healthcare, Best, The Netherlands). Global and regional brain volumes, and thicknesses of the cerebral cortex will be measured. To visualize the cerebral arteries, specifically the lumen diameter and tortuosity of all major intracranial arteries, we will acquire a high resolution, whole-brain multi-band time of flight (TOF) sequence (FOV = 200X200X120.9mm, TR/TE = 23/3.45ms, FA = 18°, parallel imaging (SENSE) acceleration factor = 3, acquisition resolution of 0.65x0.65x1.30mm, reconstructed resolution of 0.626x0.625x0.65mm). This will be followed by a single slice 2D-phase contrast velocity image that was placed using the offset of the labelling band to estimate velocity where the

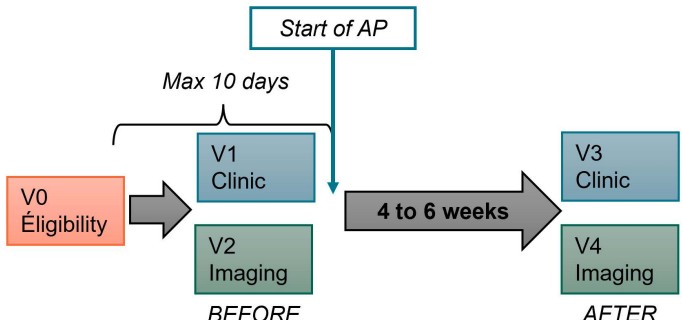

**Fig 2. Study Overview.** After the eligibility visit (V0-Eligibility) and enrollment, the participant will be invited to complete the V1-Clinic and V2-Imaging visits before starting the AP ("BEFORE"), but no later than 10 days after V0. The two other visits, V3-Clinic and V4-Imaging, will occur 4 to 6 weeks after the start of AP ("AFTER").

blood is labelled (FOV = 230x230mm; TR/TE = 9.2/6; slice thickness = 5 mm, voxel size = 0.45x0.45mm; bandwidth = 428.3 Hz/pixel; velocity encoding gradient = 80 cm/s). For structural T1-weighted images, the protocol is as follows: scan duration = 6 min, repetition time (TR) = 7.9 ms, echo time (TE) = 3.5 ms, field of view = 240 × 240 × 150, flip angle = 8 and 1mm3 isotropic voxels. The hypothalamus and pituitary gland volumes will be included because they are brain regions linked with the hypothalamus-pituitary-adrenal (HPA) stress axis, which could be hyperactive in FEP [40–42].

The Aarhus site will scan on a GE HealthCare SIGNA™ PET/MR AIR™, with the tracers, $[^{15}O]H_2O$, $[^{11}C]OHB$ and $[^{18}F]FDG$ each with simultaneous PET recording of the brain in list mode, automatic drawing of arterial blood samples, and acquisition of MR. MR sequences will include T1, T2 FLAIR and TOF images, and gamma-aminobutyric acid spectroscopy.

## Psychiatric outcome measures

The history of psychiatric diagnosis and the duration of untreated illness (calculated by subtracting the age at which the first symptoms of psychosis is reported from the age at first meeting with the FEP team) will be assessed by the study psychiatrist based on DSM-V-TR or ICD-10 criteria. Validated questionnaires will be used including the Brief Psychiatric Rating Scale to assess psychotic symptoms [43] in Sherbrooke and the Positive and Negative Syndrome Scale [44] in Aarhus, the Calgary Depression Scale for Schizophrenia for depressive symptoms [45], the Brief Cognitive Assessment in Schizophrenia [46], the Clinical Global Impressions Severity Scale for global severity of symptoms [47], the Global Assessment of Functioning [48], and the Alcohol Use Disorders Identification Test and Drug Use Disorders Identification Test, and Fagerström Test for drug, alcohol and nicotine dependences [49]. Sleep disorder will be assessed by the Epsworth Sleepiness Scale (ESS). Side effects of AP will be measured by the Side Effect Rating Scale (SERS-UKU) and the Barnes Akathysia Scale (BAS). The dose of AP will be adjusted with chlorpromazine equivalents [50]. Adherence to AP will be assessed with the Medication Adherence Rating Scale [51]. Exercise will be measured as average weekly hours of sport/exercise per day as measured by question 4 of the Simple Physical Activity questionnaire SIMPAQ (hour). Medication type and dose will be tracked.

## Clinical and laboratory measures

**Anthropometric biomarkers.** Weight, height, body mass index (BMI), waist circumference, blood pressure and pulse will be measured at V1 and V3.

**General, metabolic and inflammation biomarkers.** The general medical assessment will include a complete blood count, renal function (creatinine), liver function (AST, albumin), thyroid function (TSH), and pregnancy test (β-HCG) for

women. The plasma metabolic profile will include fasting glucose, insulin, and hemoglobin A1c, triglycerides, cholesterol, lactate, and ketones, plus metabolomic and inflammatory profiles. Plasma caprylic and capric acids will be measured by gas chromatography-mass spectrometry (GC-MS; Agilent, Waldbronn, Germany). Glucose levels will be monitored from CGM as mean±SD and time in range. Inflammatory biomarkers will include C-reactive protein, interleukins-6 and 2, and tumor necrosis factor-alpha.

## Data analysis

**PET data analysis.** PET tracer kinetics will be assessed by the Patlak method to quantify the brain uptake rate constants ($K_{glu}$, $K_{AcAc}$ [Sherbrooke]¸or $K_{OHB}$ [Aarhus]; $min^{-1}$) and cerebral metabolic rate (CMRAcAc/OHB; CMRGlu; µmol/100 g/min) [52].

**MRI data analysis.** Regional and whole brain volumes and cortical thicknesses will be determined using FreeSurfer Suite 6.0 or newer (Martinos Center for Biomedical Imaging, Cambridge, MA) [17]. Regional volumes will be normalized to the intracranial volume of each participant [53].

## Statistical analysis

At both sites, pre-post AP differences will be evaluated using Wilcoxon signed-rank non-parametric tests for paired data. Correlations will also be assessed between brain function (variables related to global and regional changes in brain energy metabolism) and blood levels of the various markers, as well as scores from different clinical questionnaires (global psychopathology, functioning, depression, sleep, and cognition). We will also assess normality for each variable and choose parametric or non-parametric tests accordingly. Summary statistics will be adapted to the data distribution, using means/SD or medians/IQR as appropriate.

Data will be presented as mean ±SD. All statistical analyses will be performed using SPSS 25.0 software (SPSS Inc., Chicago, Illinois, USA) using two-sided tests with statistical significance defined as $p < 0.05$. Each site will independently compile and analyze scans and outcomes. In an additional exploratory step, federated learning will be used to further analyze and compare the two datasets in greater depth. Federated learning allows analysis of multiple datasets located on different devices, without exchanging and sharing raw data [54].

## Discussion

### Strengths and limitations

One of the strengths of the present study is the target population, FEP, which is a transdiagnostic population because patients attending a FEP clinic end up with a variety of diagnoses ranging from psychotic disorder to mood disorder, substance use disorder and personality disorder [55,56]. That heterogeneity implies that our study results could potentially be extrapolated broadly across different forms of mental illness. The more homogeneous sample in Aarhus will help determine whether the early stage of schizophrenia is metabolically similar to that of psychosis more broadly defined (Sherbrooke). Additionally, a broad spectrum of recruitment criteria with few exclusions will make the results more generalizable.

This study is observational and will thus have no direct influence on clinical practice. It is expected that a total of n=24 participants will complete the study on the two sites combined. This should provide ample power to define the metabolic perturbations in the brain. However, the heterogeneity in diagnosis and AP treatment will limit the power to extrapolate the findings. Given the small sample size and the exploratory nature of this proof-of-concept study, statistical significance should be interpreted with caution and the results should be viewed as preliminary and intended to inform future research. Participants will be included only if the AP can be delayed by a maximum of 10 days, and if they agree to receive an AP. Thus, symptoms of enrolled participants could be less severe than usual. Lifestyle factors (e.g., diet, exercise, substance

use) could influence brain energy metabolism and metabolic markers but are not a focus of this study. Although a high dropout rate is anticipated in this population, our recruitment strategy accounts for this. However, if attrition is higher than expected, it may reduce power and introduce selection bias. In Sherbrooke, limited funding means we will rely on an existing database of healthy controls rather than enrolling a matched control group undergoing the same imaging and metabolic assessments concurrently. This could introduce bias due to differences in data collection methods and effects of timeline. However, this bias will be reduced and controlled by the use of healthy controls (matched on BMI, sex, and parental, social and economic status) in Aarhus. While a focus on FEP is a strength in some respects, it limits applicability to individuals with chronic psychosis or bipolar disorders or those who have been on long-term AP treatment.

## Perspective

This will be the first proof of concept study to assess whether the brain energy defect in FEP is specific to glucose. If confirmed, it will reinforce the rationale for therapeutic ketogenic interventions (diet or exogenous or mildly ketogenic drugs) as a potential treatment for FEP, psychosis and mood spectrum disorders. Several research and hypothesis papers already link disordered brain energy metabolism to development of mental disorders, collectively strengthening the concept of common pathophysiology of mental disorders being linked to metabolic dysregulation. This will help move the field beyond a focus limited to neurotransmitters models, i.e., dopamine for psychosis [57].

A secondary question that will await after this study is that if brain ketone metabolism is still intact in FEP, should ketogenic interventions be used to try to correct the brain glucose deficit? If so, from a practical perspective, exogenous ketones should be considered in FEP because most of these patients are unlikely to maintain a strict ketogenic diet, which is needed to attain plasma ketone levels consistently higher than 0.5 mmol/L. In fact, in the absence of institutionalization [22], the cognitive and negative symptoms of FEP are a significant obstacle to the chances of successfully implementing a strict ketogenic diet in FEP in their home and routines. Exogenous ketones are typically available in supplement form, i.e., medium chain triglyceride, ketone salt, or ketone ester [58], and can provide an alternative energy source for the brain by increasing circulating ketone levels without necessarily restricting dietary carbohydrates. They are safe and well tolerated over long periods [18,59], so could be used as an adjunctive therapy to AP. Concomitant intermittent fasting, and/or medications that promote ketosis could also be beneficial [60,61] but these approaches require study in patients.

Finally, this study may inform biological biomarkers that could be a proxy of impaired brain glucose metabolism (i.e., CGM, lactate levels), which could represent a non-invasive pragmatic way to identify patients who may most benefit from ketogenic interventions.

## Supporting information

**S1 File  Original procol.**
(PDF)

**S2 File.  SPIRIT checklist.**
(DOCX)

## Author contributions

**Conceptualization:** Kevin Zemmour, Mélanie Fortier, Allan Kjeldsen Hansen, Margaret Hahn, Stephen Cunnane.

**Funding acquisition:** Stephen Cunnane.

**Methodology:** Kevin Zemmour, Mélanie Fortier, Allan Kjeldsen Hansen, Margaret Hahn, Stephen Cunnane.

**Writing – original draft:** Kevin Zemmour, Guy-Olivier Samson, Mélanie Fortier, Eliot Parent, Bjørn Hylsebeck Ebdrup, Margaret Hahn, Stephen Cunnane.

**Writing – review & editing:** Kevin Zemmour, Guy-Olivier Samson, Mélanie Fortier, Alexandra Leus, Sylvain Grignon, Jean-Daniel Carrier, Kevin Whittingstall, Andreas Aalkjær Danielsen, Ole Köhler-Forsberg, Allan Kjeldsen Hansen, Sri Mahavir Agarwal, Anna Cristina Andreazza, Bjørn Hylsebeck Ebdrup, Margaret Hahn, Stephen Cunnane.

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
