## [Decision Letter · Decision Letter 0]

31 Mar 2025

PONE-D-25-06878Brain glucose and ketone metabolism in first-episode psychosis: neuroimaging and brain metabolism before and after antipsychotic treatment: the protocol for the CAST-ATP studyPLOS ONE

Dear Dr. Hahn,

Thank you for submitting your manuscript to PLOS ONE. After careful consideration, we feel that it has merit but does not fully meet PLOS ONE’s publication criteria as it currently stands. Therefore, we invite you to submit a revised version of the manuscript that addresses the points raised during the review process. Specifically reviews requested additional information regarding the sample size and distribution, and how this influenced the statistics.

We look forward to receiving your revised manuscript.

Kind regards,

Colin Johnson, Ph.D.

Academic Editor

PLOS ONE

Journal Requirements:

Financial support for this article and the project it describes was received in the form of a philanthropic grant donated by the “Baszucki Brain Research Fund”, USA, to the Université of Sherbrooke Foundation. SCC holds the Endowed Clinical Ketotherapeutics Chair at the Université of Sherbrooke. MH holds the CAMH and UofT Meighen Family Research Chair in Psychosis Prevention. SMA is supported in part by the Discovery Fund, CAMH and the Academic Scholars Awrard from the Department of Psychiatry, University of Toronto.

SCC has received research funding and test materials from Nestlé Health Science. He consults for Nestlé Health Science and Cerecin. OKF reports speaker fees from Lundbeck Pharma A/S, consultant work for WCG International, and serving on an advisory board for Boehringer-Ingelheim. BE is part of the Advisory Board of Boehringer Ingelheim, Lundbeck Pharma A/S; and has received lecture fees from Boehringer Ingelheim, Otsuka Pharma Scandinavia AB, and Lundbeck Pharma A/S. SMA has received honoraria from HLS therapeutics and Boehringer Ingelheim. MH has received consultant fees from Alkermes and MERCK. The other authors declare that the research will be conducted in the absence of any commercial or financial relationships that could be construed as a potential conflict of interest.

6. Please provide a complete Data Availability Statement in the submission form, ensuring you include all necessary access information or a reason for why you are unable to make your data freely accessible. If your research concerns only data provided within your submission, please write "All data are in the manuscript and/or supporting information files" as your Data Availability Statement.

Reviewers' comments:

Reviewer's Responses to Questions

**Comments to the Author**

1. Does the manuscript provide a valid rationale for the proposed study, with clearly identified and justified research questions?

Reviewer #1: Yes

Reviewer #2: Yes

2. Is the protocol technically sound and planned in a manner that will lead to a meaningful outcome and allow testing the stated hypotheses?

Reviewer #1: Yes

Reviewer #2: Yes

3. Is the methodology feasible and described in sufficient detail to allow the work to be replicable?

Reviewer #1: Yes

Reviewer #2: Yes

4. Have the authors described where all data underlying the findings will be made available when the study is complete?

Reviewer #1: Yes

Reviewer #2: Yes

5. Is the manuscript presented in an intelligible fashion and written in standard English?

Reviewer #1: Yes

Reviewer #2: Yes

6. Review Comments to the Author

You may also provide optional suggestions and comments to authors that they might find helpful in planning their study.

Reviewer #1: The objectives (primary, secondary and exploratory) are outlined appropriately and the document is well written. There are a number of clarifications needed as noted below.

As per the investigators, this study is observational and will thus have no direct influence on clinical practice. It is expected that a total of n=24 participants will be needed to complete the study on the two sites combined. The investigators note this should provide ample power to define the metabolic perturbations in the brain. How is power being used here?

There are many analyses being proposed for so few patients. One needs to exercise caution on how to interpret the p-values as this is noted as a proof of concept study as per the investigators. This should be noted as a limitation.

The sample size rationale appears reasonable as there is no background data to establish it with any power. However, could some clinically meaningful effect sizes for differences be used to establish the sample size? In any case, be specific. If you need 24 subjects from the two sites is that 12 per group (AP vs. controls)? Also what about dropouts? How will they be handled in this short period if that should be an issue? This should be added as a possible limitation if such is the case.

The statistical analysis approach for this limited sample is reasonable. However, if the investigators are proposing a non parametric approach such as the Wilcoxon signed rank for paired data then one may wish to consider the IQR (interquartile range) and not only the mean and standard deviation for summary results if some data satisfies the normality assumption. There are a number of outcome measures. Specific analyses should be matched with those being proposed. The timeline in Table 1 was very helpful.

Reviewer #2: The protocol’s hypothesis that antipsychotic treatment may worsen brain glucose metabolism, while leaving ketone metabolism intact, is particularly relevant to advancing therapeutic strategies. Another strength is that this protocol is novel in that direct neuroimaging measurement of both glucose and ketone metabolism before and after antipsychotic initiation in FEB is, I believe as you stated, unprecedented.

You have clearly referenced recent metabolic psychiatry and ketogenic diet studies related to serious mental illnesses (Lines 77-89), briefly provided examples of existing PET studies (Lines 94-96) and explicitly mentioned the absence of direct measurement of brain glucose and ketone metabolism in FEP (Lines 90-92). A possible improvement would be to add a short paragraph summarizing recent PET imaging or relevant metabolic intervention studies in psychosis if they exist or are underway in clinical trials. This would strengthen the claim making the claim of novelty explicit, verifiable and contextualized clearly within existing research, but only as manuscript length allows. What has already been provided in the manuscript is sufficient. This could be reserved for subsequent publications reporting empirical results from this protocol.

I appreciate that one of the exclusion criteria is “psychosis induced by drug use or withdrawal” and I like that you state clearly a psychiatrist will make that determination through review of patient file, psychiatric interview and first eligibility visit by a researcher team member. I think that this may still be difficult to determine. Will any standardized structured interviews or special clinical instruments be used to help determine this? Will you rely on or conduct toxicology screenings to rule out recent drug use. How will it be determined in cases of chronic drug use leading up to first psychosis? It may help other researchers using this protocol to have that level of detail from your work.

The Aarhus site explicitly mentions having access to MRS to measure GABA. Given the significant metabolic relevance of glutamate, and emerging evidence suggesting substantial glutamatergic reductions with other metabolic interventions (e.g., ketogenic diets), I suggest considering measuring brain glutamate levels via MRS at sites with available equipment. Or possibly peripheral biomarkers reflecting glutamatergic metabolism could also be explored, if feasible. Including glutamate measurements would allow valuable comparison with metabolic changes seen in other interventions and disorders, potentially clarifying how antipsychotic initiation uniquely affects glutamate metabolism in first-episode psychosis. I do recognize, however, that funding or logistical constraints may not allow for such an expansion; I suggest this merely because it could provide interesting insights.

In regards to secondary and exploratory objectives, these are well aligned. I wonder if the use of a standardized quality of life measure as used in other metabolic psychiatry studies would be well-aligned and enrich the interpretation of the primary outcomes.

Given that sleep disturbances are both common in first episode psychosis and frequently affected (either positively or negatively) by antipsychotic treatment, sleep disruption could represent a confounding variable as you are studying metabolism in this population. I am wondering if explicitly clarifying how you will evaluate this in your statistical analysis plan would strengthen the protocol. Or a mention of whether or how you intend to account analytically for variations in sleep disruption (as measured by the Epworth Sleepiness Scale).

Wilcoxon is clearly justified in the text and appropriate given the sample size and repeated-measures design. Correlational analysis is appropriate, straightforward and adequately described. I particularly like the proposed use of Federated Learning (as described in lines 287-289) as an ethical and statistical choice in studying this vulnerable population across international sites.

Overall this protocol is very well done, methodologically sound, extremely clean and professional. That said, I found just a few minor punctuation and formatting points.

Line 95: "… aripiprazole and ziprasidone are reported be associated …" Correction: should be "are reported to be associated" (missing the word "to").

Lines 164-165: "The exception will be aripiprazole if it taken at less than 2.5 mg/day…" Correction: should be "if it is taken…" (missing the word "is").

Line 249: "Side effects of AP will be measured by the (i) Side Effect Rating Scale (SERS-UKU), and the Barnes Akathysia Scale (BAS)." Correction: No comma needed before "and" here since only two items are listed. Better phrasing: "...measured by the Side Effect Rating Scale (SERS-UKU) and the Barnes Akathisia Scale (BAS)."

Line 357: "Academic Scholars Awrard" Correction: should be "Academic Scholars Award" (typo).

7. PLOS authors have the option to publish the peer review history of their article (what does this mean? ). If published, this will include your full peer review and any attached files.

**Do you want your identity to be public for this peer review?** For information about this choice, including consent withdrawal, please see our Privacy Policy .

Reviewer #1: No

Reviewer #2: **Yes: ** Nicole Laurent

---

## [Author Response · Author response to Decision Letter 1]

3 May 2025

Comments to authors

The revisions were very helpful to improve the article. Find modifications and answers to reviewers. Text in red in the manuscript is revised from the original.

Reviewer #1:

The objectives (primary, secondary and exploratory) are outlined appropriately and the document is well written. There are a number of clarifications needed as noted below.

1) As per the investigators, this study is observational and will thus have no direct influence on clinical practice. It is expected that a total of n=24 participants will be needed to complete the study on the two sites combined. The investigators note this should provide ample power to define the metabolic perturbations in the brain. How is power being used here?

RESPONSE: “Ample power” was not meant to indicate a formal power calculation, but rather to reflect our expectation that a sample of 24 is expected be sufficient to detect a meaningful difference in global and regional brain ketone uptake versus brain glucose uptake in FEP. Given that this patient population is totally new in terms of brain glucose and ketone PET imaging, this study is powered to identify clear signals to guide future research.

2) There are many analyses being proposed for so few patients. One needs to exercise caution on how to interpret the p-values as this is noted as a proof of concept study as per the investigators. This should be noted as a limitation.

RESPONSE: We agree that, given the small sample size and exploratory nature of the study, p-values must be interpreted with caution. This will be clearly acknowledged as a limitation (change made on page 9).

3) The sample size rationale appears reasonable as there is no background data to establish it with any power. However, could some clinically meaningful effect sizes for differences be used to establish the sample size? In any case, be specific. If you need 24 subjects from the two sites is that 12 per group (AP vs. controls)?

RESPONSE: The sample size estimate is based on brain glucose and ketone uptake values obtained from over 400 PET scans across a wide range of ages, clinical conditions, and interventions. While no study to date has investigated this specific population—individuals with a first episode of psychosis (FEP)—using this metabolic imaging approach, our study is exploratory in nature and aims to provide foundational data to guide future, larger trials. Given the expected heterogeneity among FEP participants, we are confident that a sample size of n=24 across both sites (approximately 12 per group) will be sufficient to establish feasibility and detect meaningful patterns in brain fuel metabolism that may inform future effect size estimations and study designs.

4) Also what about dropouts? How will they be handled in this short period if that should be an issue? This should be added as a possible limitation if such is the case.

RESPONSE: Analyses will be based on complete data (before and after AP use). In the event of higher-than-expected dropout, we will assess the characteristics of participants lost to follow-up and may explore additional sensitivity or descriptive analyses as appropriate. Also, we have already accounted for potential important attrition by planning to recruit up to 18 FEP participants per site with the goal of obtaining complete data on at least 10 of them. However, if attrition exceeds expectations, it may further reduce statistical power and introduce selection bias, particularly if dropouts differ systematically from completer This will be added as a limitation (changes made page 9).

5) The statistical analysis approach for this limited sample is reasonable. However, if the investigators are proposing a non parametric approach such as the Wilcoxon signed rank for paired data then one may wish to consider the IQR (interquartile range) and not only the mean and standard deviation for summary results if some data satisfies the normality assumption.

RESPONSE: Given our small sample size, we agree that a more comprehensive analytical approach could be develop. We will assess normality for each variable and choose parametric or non-parametric tests accordingly. Summary statistics will be adapted to the data distribution, using means/SD or medians/IQR as appropriate (change made page 8).

6) There are a number of outcome measures. Specific analyses should be matched with those being proposed. The timeline in Table 1 was very helpful

RESPONSE: We have revised the manuscript to ensure consistency between the proposed analyses and the corresponding outcome measures (page 8).

Reviewer #2:

1) The protocol’s hypothesis that antipsychotic treatment may worsen brain glucose metabolism, while leaving ketone metabolism intact, is particularly relevant to advancing therapeutic strategies. Another strength is that this protocol is novel in that direct neuroimaging measurement of both glucose and ketone metabolism before and after antipsychotic initiation in FEB is, I believe as you stated, unprecedented. You have clearly referenced recent metabolic psychiatry and ketogenic diet studies related to serious mental illnesses (Lines 77-89), briefly provided examples of existing PET studies (Lines 94-96) and explicitly mentioned the absence of direct measurement of brain glucose and ketone metabolism in FEP (Lines 90-92). A possible improvement would be to add a short paragraph summarizing recent PET imaging or relevant metabolic intervention studies in psychosis if they exist or are underway in clinical trials. This would strengthen the claim making the claim of novelty explicit, verifiable and contextualized clearly within existing research, but only as manuscript length allows. What has already been provided in the manuscript is sufficient. This could be reserved for subsequent publications reporting empirical results from this protocol.

RESPONSE: We are confident in the statement that no measurements of brain ketone metabolism (PET or MRS) have been reported. We agree that summarizing these emerging studies and relevant metabolic interventions in psychosis could further contextualize our protocol and support the claim of novelty. However, given current manuscript length constraints, we will reserve this for future publications reporting empirical findings from this study.

2) I appreciate that one of the exclusion criteria is “psychosis induced by drug use or withdrawal” and I like that you state clearly a psychiatrist will make that determination through review of patient file, psychiatric interview and first eligibility visit by a researcher team member. I think that this may still be difficult to determine. Will any standardized structured interviews or special clinical instruments be used to help determine this? Will you rely on or conduct toxicology screenings to rule out recent drug use. How will it be determined in cases of chronic drug use leading up to first psychosis? It may help other researchers using this protocol to have that level of detail from your work.

RESPONSE: Participants will be patients of our two FEP clinics. “Simple, short” psychosis induced by drug use or withdrawal is an exclusion criterion. However, in the case of chronic use of drug, psychotics symptoms can be longer and may mimic primary psychosis. In that case, where it’s unclear if psychotics symptoms are primary or secondary, such patients may end up being included in this study. Comorbidities will be written up for these participants. This will be noted as a limitation when the results are submitted for publication.

3) The Aarhus site explicitly mentions having access to MRS to measure GABA. Given the significant metabolic relevance of glutamate, and emerging evidence suggesting substantial glutamatergic reductions with other metabolic interventions (e.g., ketogenic diets), I suggest considering measuring brain glutamate levels via MRS at sites with available equipment. Or possibly peripheral biomarkers reflecting glutamatergic metabolism could also be explored, if feasible. Including glutamate measurements would allow valuable comparison with metabolic changes seen in other interventions and disorders, potentially clarifying how antipsychotic initiation uniquely affects glutamate metabolism in first-episode psychosis. I do recognize, however, that funding or logistical constraints may not allow for such an expansion; I suggest this merely because it could provide interesting insights.

RESPONSE: This is a great idea and will be explored in a future study.

4) In regards to secondary and exploratory objectives, these are well aligned. I wonder if the use of a standardized quality of life measure as used in other metabolic psychiatry studies would be well-aligned and enrich the interpretation of the primary outcomes.

RESPONSE: This is a very relevant and interesting suggestion which we will certainly keep in mind for a future interventional study in development.

5) Given that sleep disturbances are both common in first episode psychosis and frequently affected (either positively or negatively) by antipsychotic treatment, sleep disruption could represent a confounding variable as you are studying metabolism in this population. I am wondering if explicitly clarifying how you will evaluate this in your statistical analysis plan would strengthen the protocol. Or a mention of whether or how you intend to account analytically for variations in sleep disruption (as measured by the Epworth Sleepiness Scale).

RESPONSE: Given our small sample size and insufficient statistical power for objective sleep measures, we will still descriptively report individual sleep scores, and we will explore potential correlations with metabolic outcomes in an exploratory manner. The statistical plan was revised to clarify our intend (page 8).

6) Wilcoxon is clearly justified in the text and appropriate given the sample size and repeated-measures design. Correlational analysis is appropriate, straightforward and adequately described. I particularly like the proposed use of Federated Learning (as described in lines 287-289) as an ethical and statistical choice in studying this vulnerable population across international sites.

7) Overall this protocol is very well done, methodologically sound, extremely clean and professional. That said, I found just a few minor punctuation and formatting points.

Line 95: "… aripiprazole and ziprasidone are reported be associated …" Correction: should be "are reported to be associated" (missing the word "to").

Lines 164-165: "The exception will be aripiprazole if it taken at less than 2.5 mg/day…" Correction: should be "if it is taken…" (missing the word "is").

Line 249: "Side effects of AP will be measured by the (i) Side Effect Rating Scale (SERS-UKU), and the Barnes Akathysia Scale (BAS)." Correction: No comma needed before "and" here since only two items are listed. Better phrasing: "...measured by the Side Effect Rating Scale (SERS-UKU) and the Barnes Akathisia Scale (BAS)."

Line 357: "Academic Scholars Awrard" Correction: should be "Academic Scholars Award" (typo).

RESPONSE: We have made the necessary corrections to spelling and punctuation errors."

Additional requirement

RESPONSE: We have ensured that the manuscript and all files comply with PLOS ONE’s style requirements.

RESPONSE: ORCID iD was add for the corresponding author.

Financial support for this article and the project it describes was received in the form of a philanthropic grant donated by the “Baszucki Brain Research Fund”, USA, to the Université of Sherbrooke Foundation. SCC holds the Endowed Clinical Ketotherapeutics Chair at the Université of Sherbrooke. MH holds the CAMH and UofT Meighen Family Research Chair in Psychosis Prevention. SMA is supported in part by the Discovery Fund, CAMH and the Academic Scholars Award from the Department of Psychiatry, University of Toronto.

RESPONSE: Statement was add in the manuscript (page 10) and the cover letter.

SCC has received research funding and test materials from Nestlé Health Science. He consults for Nestlé Health Science and Cerecin. OKF reports speaker fees from Lundbeck Pharma A/S, consultant work for WCG International, and serving on an advisory board for Boehringer-Ingelheim. BE is part of the Advisory Board of Boehringer Ingelheim, Lundbeck Pharma A/S; and has received lecture fees from Boehringer Ingelheim, Otsuka Pharma Scandinavia AB, and Lundbeck Pharma A/S. SMA has received honoraria from HLS therapeutics and Boehringer Ingelheim. MH has received consultant fees from Alkermes and MERCK. The other authors declare that the research will be conducted in the absence of any commercial or financial relationships that could be construed as a potential conflict of interest.

RESPONSE: Statement was add in the manuscript (page 10) and the cover letter.

RESPONSE: We have revised the manuscript to ensure the ethics statement appears only in the Methods section, as requested.

6. Please provide a complete Data Availability Statement in the submission form, ensuring you include all necessary access information or a reason for why you are unable to make your data freely accessible. If your research concerns only data provided within your submission, please write "All data are in the manuscript and/or supporting information files" as your Data Availability Statement.

- Je ne me rappelle pas si c’était fait. Le « submission form » doit être dans le site internet ?

RESPONSE: We have ensured that the guidelines for Supporting Information have been followed.

8. Please review your reference list to ensure that it is complete and correct. If you have cited papers that have been retracted, please include the rationale for doing so in the manuscript text, or remove these references and replace them with relevant current references. Any changes to the reference list should be mentioned in the rebuttal letter that accompanies y

---

## [Decision Letter · Decision Letter 1]

15 May 2025

Brain glucose and ketone metabolism in first-episode psychosis: neuroimaging and brain metabolism before and after antipsychotic treatment: the protocol for the CAST-ATP study

PONE-D-25-06878R1

Dear Dr. Hahn,

We’re pleased to inform you that your manuscript has been judged scientifically suitable for publication and will be formally accepted for publication once it meets all outstanding technical requirements.

Kind regards,

Colin Johnson, Ph.D.

Academic Editor

PLOS ONE

Additional Editor Comments (optional):

Reviewers' comments:

Reviewer's Responses to Questions

**Comments to the Author**

1. Does the manuscript provide a valid rationale for the proposed study, with clearly identified and justified research questions?

Reviewer #1: Yes

Reviewer #2: Yes

2. Is the protocol technically sound and planned in a manner that will lead to a meaningful outcome and allow testing the stated hypotheses?

Reviewer #1: Yes

Reviewer #2: Yes

3. Is the methodology feasible and described in sufficient detail to allow the work to be replicable?

Reviewer #1: Yes

Reviewer #2: Yes

4. Have the authors described where all data underlying the findings will be made available when the study is complete?

Reviewer #1: Yes

Reviewer #2: Yes

5. Is the manuscript presented in an intelligible fashion and written in standard English?

Reviewer #1: Yes

Reviewer #2: Yes

6. Review Comments to the Author

You may also provide optional suggestions and comments to authors that they might find helpful in planning their study.

Reviewer #1: All comments have been addressed adequately with the revisions included as needed.

As an exploratory/feasibility investigation, this will suffice.

Reviewer #2: This protocol is ready for publication. It has been well thought out and is thorough. I have no concerns it will not allow testing the stated hypotheses. It is technically sound and planned in a manner that will lead to meaningful outcomes. The writing is clear, presented in an intelligible fashion and written in standard English.

7. PLOS authors have the option to publish the peer review history of their article (what does this mean? ). If published, this will include your full peer review and any attached files.

**Do you want your identity to be public for this peer review?** For information about this choice, including consent withdrawal, please see our Privacy Policy .

Reviewer #1: No

Reviewer #2: **Yes: ** Nicole Laurent

---

## [Editor Report · Acceptance letter]

PONE-D-25-06878R1

PLOS ONE

Dear Dr. Hahn,

I'm pleased to inform you that your manuscript has been deemed suitable for publication in PLOS ONE. Congratulations! Your manuscript is now being handed over to our production team.

Kind regards,

on behalf of

Dr. Colin Johnson

Academic Editor

PLOS ONE